# Metagenomic next-generation sequencing of cell-free DNA for the identification of viruses causing central nervous system infections

Yuying Lu,[1,2] Ye Zhang,[3] Zheng Lou,[3] Xiaomin He,[3] Qinghua Zhang,[1] Qingxia Zhang,[1] Shu Zhao,[1] Han Chen,[1] Haixia Zhu,[1] Zhi Song,[1] Ruxu Zhang,[1] Caiyu Ma,[1,4] Ding Liu[1]

**ABSTRACT** Metagenomic next-generation sequencing (mNGS) can be used to detect pathogens, but there are limited data on its role in detecting viral central nervous system (CNS) infections. This was a multi-center retrospective study of patients clinically diagnosed with suspected viral CNS infection in Hunan between January 2018 and July 2021. mNGS detection was performed on cerebrospinal fluid (CSF) cell-free DNA (cfDNA) and whole-cell DNA (wcDNA) for diagnostic comparison. A total of 195 patients with suspected viral CNS infection underwent mNGS of CSF samples, and 175 of them received a final clinical diagnosis of CNS viral infection. A total of 100 (57.1%) cases were found to be virus-positive by mNGS, including 49 VZV, 32 HSV-1, 13 EBV, 5 HSV-2, and 1 porcine HSV. Compared with wcDNA, cfDNA mNGS showed a significantly lower proportion of human DNA ($n = 26$ patients; $0.91 \pm 0.1$ vs $0.96 \pm 0.08$, $P < 0.01$). Of these 26 patients, 24 were finally diagnosed with viral infections. The sensitivity of mNGS for detecting viruses using cfDNA and wcDNA was 66.7% (16/24) and 33.3% (8/24) ($P < 0.01$), respectively. Herpesviruses dominated the spectrum of DNA viruses in patients with viral CNS infections in Hunan, China, with VZV being the most common. mNGS, especially using cfDNA, is a promising complementary diagnostic method in detecting viral CNS infections.

**IMPORTANCE** This study provides significant new data on the application of metagenomic next-generation sequencing (mNGS) to clinical diagnostics of central nervous system (CNS) viral infections, which can have high mortality rates and severe sequelae. Conventional diagnostic procedures for identifying viruses can be inefficient and rely on preconceived assumptions about the pathogen, making mNGS an appealing alternative. However, the effectiveness of mNGS is affected by the presence of human DNA contamination, which can be minimized by using cell-free DNA (cfDNA) instead of whole-cell DNA (wcDNA). This multi-center retrospective study of patients with suspected viral CNS infection found that mNGS using cfDNA had a significantly lower proportion of human DNA and higher sensitivity for detecting viruses than mNGS using wcDNA. Herpesviruses, particularly VZV, were found to be the most common DNA viruses in these patients. Overall, mNGS using cfDNA is a promising complementary diagnostic method for detecting CNS viral infections.

**KEYWORDS** cell-free DNA, central nervous system infection, cerebrospinal fluid, metagenomic sequencing, herpesvirus

Viral infections of the central nervous system (CNS) have an incidence of approximately 1.5–7 per 100,000, and they are serious infections with high mortality rates and severe sequelae (1–4). Symptom severity and outcomes from CNS infections vary

Address correspondence to Ding Liu, ding.liu@csu.edu.cn, or Caiyu Ma, macaiyu421@163.com.

Y.Z., Z.L., and X.H. are employed by Hugobiotech Co., Ltd. The remaining authors declare that the research was conducted in the absence of any commercial or financial relationships that could be construed as a potential conflict of interest.

See the funding table on p. 11.

depending on the virus (5, 6), making accurate pathogen identification an essential component of quality patient management. However, virus identification can be difficult, and traditional diagnostic procedures are inefficient and pathogen-specific, requiring continuous testing of cerebrospinal fluid (CSF) and sometimes invasive surgery. Furthermore, the viral pathogen spectrum detected in CNS infections is typically complex and diverse, with potentially low viral loads. Conventional methods such as polymerase chain reaction (PCR) and antibody detection (enzyme immunoassays) often exhibit poor utility in such cases (7).

Metagenomic next-generation sequencing (mNGS) is a high-throughput sequencing technology that can be used for unbiased detection of any pathogen without the need for sequence-specific amplification (8). This technology overcomes the limitations of pathogen-specific PCR assay, thus expediting the identification of unknown causative pathogens (9). mNGS has successfully detected a range of pathogens in patients with CNS infections (10–12). Several studies have shown that mNGS can detect pathogens that were identified as negative using conventional methods (including PCR), demonstrating higher sensitivity than conventional methods (12–14). mNGS is increasingly used clinically as a powerful tool for pathogen identification (15, 16).

The effectiveness of pathogen identification through CSF mNGS is notably affected by the presence of human DNA contamination, as highlighted in prior research (17). White blood cells (WBCs), commonly found in CSF from patients with CNS infections, serve as a primary source of host-related impurities. Cell-free DNA (cfDNA) refers to completely or partially degraded endogenous DNA actively released from cells into the surrounding fluid (18). cfDNA may originate from cell lysis, apoptosis, and necrosis, and it is present in various body fluids including serum, plasma, and the CSF (19). cfDNA can be used for high-sensitivity and high-specificity pathogen detection (20), including for neuroinfectious diseases, where it has been used to detect CNS parasites, *Mycobacterium tuberculosis* (MTB), and bacteria (21). Furthermore, CSF mNGS using cfDNA can minimize the impact of human DNA contamination. However, there have been few comparisons of mNGS using cfDNA and mNGS using whole-cell DNA (wcDNA) for detecting CSF viruses.

In this study, we quantified the DNA virus spectrum in patients with suspected CNS viral infection using mNGS and analyzed the clinical characteristics of CNS infection caused by different viruses. We also compared virus detection using CSF cfDNA and wcDNA samples for mNGS.

## MATERIALS AND METHODS

### Study participants and sample collection

This retrospective, multi-center study analyzed patients with suspected CNS viral infections from 13 hospitals in Hunan, China, between January 2018 and July 2021. CSF samples were collected from eligible patients according to standard procedures and stored at −80°C in the Third Xiangya Hospital, Central South University. CSF samples used in this study were prospectively collected as part of routine clinical procedures for future analysis. To address ethical concerns associated with potential future retrospective studies using patient data and samples, we obtained patient consent during sample collection. Some samples may have been lost due to patient refusal or weekend testing, so we minimized potential bias by documenting the reasons for sample loss and evaluating differences between included and excluded cases. We gathered medical history, clinical characteristic, and ancillary investigation data.

### Inclusion and exclusion criteria

The enrolled participants comprised individuals diagnosed with suspected CNS viral infection who had not undergone PCR and antiviral antibody testing. The clinical diagnostic criteria for viral encephalitis, meningitis, meningoencephalitis, or meningomyelitis necessitate simultaneous fulfillment of the following three conditions (22): (i)

Major criteria: altered mental status, encompassing a decline in the level of consciousness, lethargy, or abnormal mental behavior persisting for ≥24 hours or the occurrence of new-onset seizures. (ii) Minor criteria: fever (body temperature ≥38 °C) either preceding or manifesting within 72 hours after onset; newly emerging focal neurological manifestations; CSF pleocytosis (>5 WBC/µL); electroencephalographic (EEG) findings indicating encephalitis; or abnormal neuroimaging suggesting encephalitis, meningitis, meningoencephalitis, or meningomyelitis. (iii) The reasonable exclusion of alternative etiologies. Patients were initially included if they satisfied both the aforementioned (i) major criteria and (ii) minor criteria. Patients were excluded if they satisfied any of the following criteria: (i) patients with microbiologically or serologically confirmed bacterial, cryptococcal, tuberculous, or parasitic CSF infections before mNGS testing or those with probable bacterial meningitis (CSF white cell polymorphs > 100 cells /µL); (ii) patients confirmed as presenting with a non-infectious CNS disease (autoimmune encephalitis, paraneoplastic syndrome, demyelinating encephalopathy, intracerebral hemorrhage, or meningeal cancer) before mNGS testing; (iii) AIDS patients; (iv) patients with incomplete data or insufficient specimen volume; and (v) patient refusal to participate.

## Routine clinical laboratory tests

Routine investigations including CSF biochemistry, CSF cytology, Gram staining, *Mycobacterium* smears, ink staining, cryptococcal capsular antigen detection, and CSF culture were performed in all patients. qPCR for HSV 1 and 2, VZV, EBV, and CMV was performed in 50 randomly selected patients using a real-time PCR Kit (Liferiver Bio-Tech, San Diego, CA).

## Metagenomic next-generation sequencing of CSF cfDNA and wcDNA

About 2 mL CSF was collected from each patient for PACEseq mNGS detection (Hugobiotech, Beijing, China). For cfDNA extraction, human cells were first removed from the CSF by centrifugation, and only the supernatant was used for subsequent extraction. For wcDNA extraction, CSF was directly used without centrifugation. The QIAamp DNA Micro Kit (Qiagen, Hilden, Germany) was applied to extract cfDNA and wcDNA according to the manufacturer's instructions. DNA libraries were then constructed using the QIAseq Ultralow Input Library Kit for Illumina (Qiagen) following the manufacturer's instructions. All constructed libraries were assessed for quality with a Qubit fluorometer (Thermo Fisher Scientific, Waltham, MA) and Agilent 2100 Bioanalyzer (Agilent Technologies, Palo Alto, CA). DNA libraries were finally sequenced on a NextSeq 550 platform (Illumina, San Diego, CA). Adapters and short, low-quality and low-complexity reads were removed from the raw data of each sequenced library. Human host DNA reads were then filtered out by alignment to the human reference database (hg38). The remaining reads were aligned to the Microbial Genome Databases (ftp://ftp.ncbi.nlm.nih.gov/genomes/). CSF cfDNA from all 195 patients and CSF wcDNA from 26 randomly selected patients were analyzed by mNGS.

## Interpretation of mNGS results

There is currently no unified standard for mNGS analysis of CSF. We established threshold criteria for positive mNGS results based on previous research on diagnosing meningitis and encephalitis using mNGS (8, 23). Virus detection was deemed positive when reads covered at least three non-overlapping genomic regions. For bacteria, fungi, and parasites, a normalized ratio using a "no template control" (NTC) sample processed in parallel was calculated to avoid false-positive detections due to reagent or laboratory contamination, and we used a reads per million (RPM) ratio (RPM-r) metric, defined as RPM-r = $RPM_{sample}/RPM_{NTC}$, with the minimum $RPM_{NTC}$ set to 1. Bacteria, fungi, or parasites were "detected" at a minimum threshold of 10 RPM-r (i.e., RPM-r ≥10) or "not detected" when RPM-r <10. mNGS for MTB was positive when at least one read was mapped to the MTB complex at either the species or genus level (24).

To determine if a detected pathogen caused the disease, two clinicians and a microbiologist evaluated the patient's clinical symptoms, laboratory results, treatment response, and outcomes. When faced with various potential diagnoses, the multidisciplinary team collaboratively arrived at a consensus to determine the most probable diagnosis. We categorized the mNGS results into four groups: (i) Confirmed Positive: where mNGS results definitively proved the presence of a neuroinvasive pathogen, consistent with the clinical diagnosis and confirmed by traditional microbiological detection methods (PCR or antibody detection) or consistent with diagnosis of clinical experience, and no other CNS diseases were evident during follow-up, (ii) Suspected Positive: where mNGS detected pathogens potentially related to the disease, which could be latent infection, viral carrier status, active infections, or a potential new, rare, or unexpected pathogen, requiring repeated tests or further confirmation through multiple microbiological methods. (iii) Inconsistent with Clinical Diagnosis: where the mNGS results presented a pathogen that did not align with the clinical diagnosis or expected disease profile. (iv) Causative Pathogens Not Detected: where the mNGS results did not identify any pathogen responsible for the disease.

## Statistical analysis

SPSS v18.0 (IBM Statistics, Armonk, NY) was used for data analysis. Measured data were described by means ± SD or medians (interquartile range; IQR), and groups were compared with the independent samples $t$-test when data were normally distributed or with the Mann–Whitney test otherwise. Count data are presented as ratios and were analyzed by the chi-squared test. $P$ values < 0.05 were considered statistically significant.

## RESULTS

### Clinical data

A total of 497 patients with suspected CNS infection were initially included, and 195 patients with suspected CNS viral infection were finally included for submission of CSF samples for mNGS. A flow chart of included and excluded patients is shown in Fig. 1.

The mean age of the 195 patients (65.6% male) was 41.5 ± 16.9 years, including 20 (10.3%) under 18 years of age. The cohort consisted mainly of patients with isolated meningitis (93, 47.7%) or encephalitis (37, 19.0%). The final diagnoses included 175 viral CNS infections, 8 tuberculous CNS infections, 5 bacterial meningitis cases, and 7 non-infectious CNS cases. Among the 50 patients randomly selected for qPCR testing (48 with a final diagnosis of CNS viral infection and 2 with non-viral infections), only 11 cases (8 VZV, 1 HSV-1, and 2 EBV) were found positive by qPCR. Of these, nine cases concurred with the mNGS results, while two discorded.

### mNGS results of CSF

In the mNGS results for 175 patients ultimately diagnosed with CNS viral infections (Fig. 2), 100 cases (57.1%) were identified as "Confirmed Positive" with pathogens, including 49 VZV, 32 HSV-1, 13 EBV, 5 HSV-2, and 1 porcine HSV, all determined as single-pathogen infections. Although 14 cases initially showed mixed-virus detection by mNGS, further clinical and diagnostic criteria categorized them as single-virus infections. Among the "Confirmed Positive" patients, the median number of mNGS reads was 345 (IQR: 24–4,859). Nine patients (5.1%) with detected pathogens were considered "Suspected Positive," including six with EBV (reads 3–17) and three with torque teno virus (TTV) (reads 3–6). Among these, three patients had positive serum EBV IgG but negative IgM, two patients had a history of lymphoproliferative disorders, and one patient had infectious mononucleosis 3 months before admission. TTV detection was considered unexpected. In six patients diagnosed with viral encephalitis (meningitis), MTB was detected (reads 1–2), while no causative viruses were detected, categorizing it as "Inconsistent with Clinical Diagnosis." In 60 cases (34.3%), no causative pathogen was detected, falling into the category of "Causative Pathogens Not Detected."

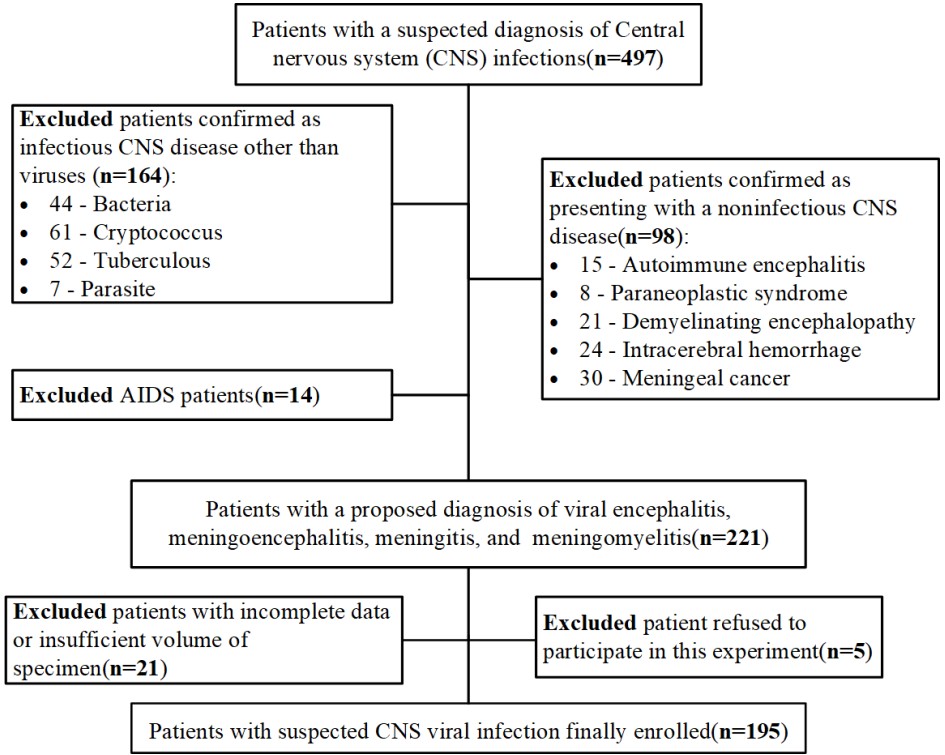

**FIG 1** Flow chart of the inclusion and exclusion of patients in the screened cohort.

In 20 cases ultimately diagnosed as non-CNS viral infections, 8 cases were diagnosed as CNS tuberculosis infection, with only 1 case detecting the MTB sequence through mNGS; 5 cases were diagnosed as bacterial meningitis, with 1 case detecting the brucella sequence; 7 cases were diagnosed as non-infectious CNS disorders, among which the

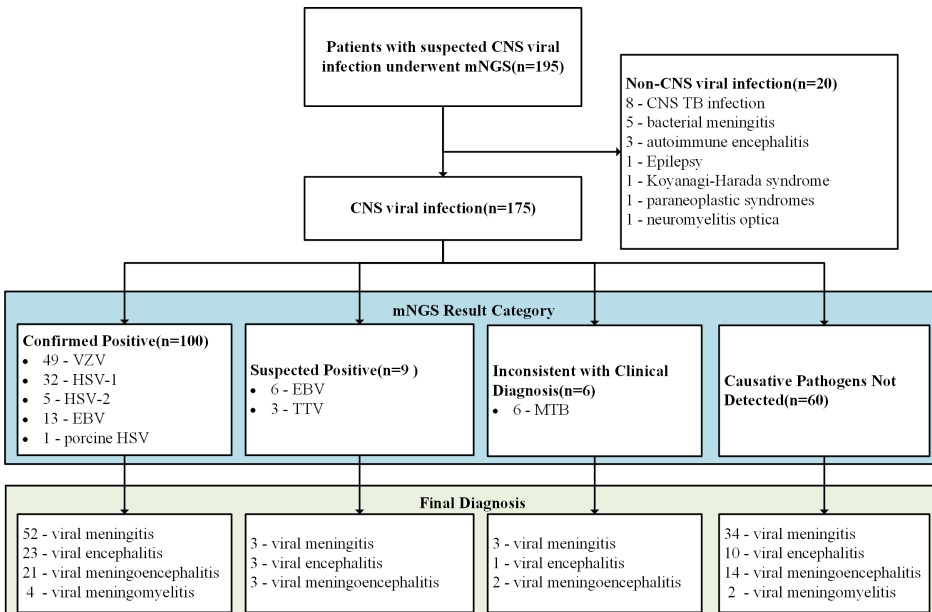

**FIG 2** Metagenomic next-generation sequencing (mNGS) of the CSF samples of 175 patients with CNS viral infections. Abbreviations: CNS, central nervous system; TB, tuberculous; HSV, herpes simplex virus; VZV, varicella zoster virus; EBV, Epstein–Barr virus; TTV, torque teno virus; MTB, Mycobacterium tuberculosis.

HSV-1 sequence was detected in 1 patient diagnosed with Koyanagi–Harada syndrome, while no pathogens were detected in the remaining 6 patients.

## Comparison of the characteristics of patients with HSV-1, VZV, and EBV infection

VZV (28.0%) was the most common virus detected by mNGS in 175 patients with CNS viral infections, followed by HSV-1 (18.3%) and EBV (7.4%). We compared the characteristics of patients with these three common viruses (Table 1). Although the causes of infection varied, the clinical presentation of these patients was similar. With respect to the laboratory data, CSF WBC counts were significantly higher in patients with VZV (median, IQR: 262, 120–462) than with HSV-1 (76.0, 51.25–151.5) or EBV (104, 45.5–229) infection, and the VZV read count (1,551, 49–6,542) detected by mNGS was also higher than in the HSV-1 (19.5, 9.25–1,411) and EBV (35, 16–84) read counts ($P < 0.05$, respectively). Patients with HSV-1 (692, 519–985.5) infection had a significantly lower CSF protein level than those with VZV (1,034, 854–1,639) and EBV (1,262, 911–1,576) infections ($P < 0.05$). There was also a statistically significant difference in the imaging findings of VZV and HSV-1 cases, with more meningeal involvement in VZV infection

**TABLE 1** Comparison of clinical and laboratory data in patients with VZV, HSV-1, and EBV virus infections[a]

| | VZV | HSV-1 | EBV | P value | | |
| | n = 49 (a) | n = 32 (b) | n = 13 (c) | A vs B | A vs C | B vs C |
|---|---|---|---|---|---|---|
| Demographic and clinical data | | | | | | |
| Age (y) | 44.29 ± 17.60 | 42.47 ± 19.93 | 49.15 ± 12.41 | 0.660 | 0.354 | 0.248 |
| Gender (male) (n, %) | 30 (61.2%) | 19 (59.4%) | 8 (61.5%) | 1.000 | 1.000 | 1.000 |
| Fever (n, %) | 29 (59.2%) | 27 (84.4%) | 12 (92.3%) | 0.026 | 0.045 | 0.656 |
| Headache (n, %) | 40 (81.6%) | 29 (90.6%) | 12 (92.3%) | 0.347 | 0.673 | 1.000 |
| Vomiting (n, %) | 23 (46.9%) | 13 (40.6%) | 3 (23.1%) | 0.651 | 0.205 | 0.322 |
| Symptom duration | 7.04 ± 5.18 | 7.04 ± 4.64 | 12.92 ± 10.71 | 0.991 | 0.011 | 0.023 |
| Seizure (n, %) | 4 (8.2%) | 5 (15.6%) | 3 (23.1%) | 0.471 | 0.153 | 0.672 |
| Cognitive disorder (n, %) | 2 (4.1%) | 6 (18.9%) | 2 (15.4%) | 0.053 | 0.145 | 1.000 |
| Speech disorder (n, %) | 6 (12.2%) | 7 (21.9%) | 2 (15.4%) | 0.354 | 0.670 | 1.000 |
| Dyskinesia (n, %) | 7 (14.3%) | 5 (15.6%) | 3 (23.1%) | 1.000 | 0.424 | 0.672 |
| Mental disorder (n, %) | 4 (8.2%) | 10 (31.3%) | 4 (30.8%) | 0.014 | 0.027 | 1.000 |
| Altered consciousness (n, %) | 6 (12.2%) | 9 (28.1%) | 4 (30.8%) | 0.086 | 0.196 | 1.000 |
| GCS | 14.61 ± 0.98 | 13.81 ± 1.47 | 13.54 ± 2.72 | 0.004 | 0.025 | 0.664 |
| Neck stiffness (n, %) | 29 (59.2%) | 15 (46.9%) | 8 (61.5%) | 0.262 | 1.000 | 0.514 |
| Cranial nerve palsies (n, %) | 8 (16.3%) | 2 (9.4%) | 3 (8.6%) | 0.301 | 1.000 | 0.567 |
| Hospital days | 16.82 ± 10.94 | 18.34 ± 7.54 | 24.38 ± 14.08 | 0.493 | 0.041 | 0.068 |
| Laboratory data (mean ± SD/median ± IQR) | | | | | | |
| CSF WBC count ($10^6$ /L) | 262 (120–462) | 76.0 (51.25–151.5) | 104 (45.5–229) | <0.001 | 0.023 | 0.841 |
| CSF neutrophil proportion (%) | 3.06 ± 10.26 | 2.72 ± 6.36 | 3.03 ± 10.83 | 0.866 | 0.695 | 0.669 |
| CSF lymphocyte proportion (%) | 81.43 ± 13.04 | 82.61 ± 12.24 | 85.04 ± 12.50 | 0.684 | 0.144 | 0.229 |
| CSF protein (mg/L) | 1034 (854–1639) | 692 (519–985.5) | 1262 (911–1576) | 0.001 | 0.416 | 0.001 |
| CSF/serum glucose (mmol/L) | 0.60 ± 0.17 | 0.61 ± 0.14 | 0.44 ± 0.19 | 0.776 | 0.007 | 0.003 |
| CSF chlorine (mmol/L) | 119.89 ± 6.59 | 121.73 ± 4.80 | 118.48 ± 6.61 | 0.178 | 0.499 | 0.073 |
| Blood leukocyte count ($10^9$ /L) | 7.63 ± 2.40 | 8.33 ± 2.82 | 7.47 ± 3.36 | 0.234 | 0.511 | 920 |
| Serum sodium (mmol/L) | 137.74 ± 4.25 | 135.96 ± 5.04 | 136.38 ± 6.49 | 0.092 | 0.496 | 0.120 |
| mNGS read number | 1551 (49–6542) | 19.5 (9.25–1411) | 35 (16–84) | 0.033 | 0.048 | 0.841 |
| Cerebral imaging characteristics (n, %) | | | | | | |
| Meningitis | 39 (79.6%) | 4 (12.5%) | 6 (46.1%) | <0.001 | 0.527 | 0.022 |
| Encephalitis | 3 (6.1%) | 13 (40.6%) | 2 (15.4%) | <0.001 | 0.280 | 0.165 |
| Meningoencephalitis | 6 (6.7%) | 12 (37.5%) | 4 (30.8%) | 0.013 | 0.196 | 0.743 |
| Meningomyelitis | 1 (2.0%) | 3 (9.4%) | 1 (7.7%) | 0.295 | 0.378 | 1.000 |

[a]HSV, herpes simplex virus; VZV, varicella zoster virus; EBV, Epstein–Barr virus; CSF, cerebrospinal fluid; GCS, Glasgow Coma Scale.

(79.6%) than in HSV-1 infection (12.5%), and parenchymal brain involvement was more frequent in HSV (40.6%) than in VZV (6.1%) infection ($P < 0.05$).

There was a positive correlation between WBC counts and detected reads of pathogenic viruses by mNGS (Fig. 3a), with the viral read count by mNGS increasing with increasing CSF WBC count. In the cohort of 100 patients with confirmed positive virus detection, 18 patients (13 VZV, 4 HSV-1, and 1 HSV-2) had CSF WBC counts over $400 \times 10^6$ /L. Of these 13 patients with VZV, nine had >2,000 reads. We also compared the mNGS detection rates between those diagnosed with viral encephalitis, meningitis, meningoencephalitis, and meningomyelitis (Fig. 3b). There were no significant differences in detection rates according to clinical symptoms or laboratory parameters.

## Comparison of cfDNA and wcDNA results

Both cfDNA and wcDNA mNGS were used to detect pathogens in 26 patients. Compared with wcDNA, cfDNA mNGS detected a significantly lower proportion of human DNA (0.91 ± 0.1 vs 0.96 ± 0.08, $P < 0.01$). The ratio of non-human DNA (microbial DNA) detected by cfDNA to that detected by wcDNA is shown in Fig. 4a. Twenty of 26 cases (76.9%) had more microbial DNA detected in cfDNA than in wcDNA, with ratios ranging from 1.0 to 24.1. Eighteen (69.2%) contained more than double the microbial DNA in cfDNA than in wcDNA. Of the seven cases with more microbial DNA detected by wcDNA, only one showed more than twice the microbial DNA in wcDNA samples.

Of these 26 patients, 24 were finally diagnosed with viral infection. The sensitivity of mNGS for detecting viruses using cfDNA and wcDNA was 66.7% (16/24) and 33.3% (8/24), respectively. Among the eight cases diagnosed with viral infection and with positive wcDNA, all were also cfDNA-positive, and the same viruses were detected in seven out of the eight cases (Fig. 4b). The read numbers and RPM of the shared viruses in cfDNA were significantly higher than those in wcDNA (5,359 ± 10,374 vs 138 ± 291 and 12.65 ± 24.53 vs 1,035.91 ± 2,278.83, respectively, $P < 0.01$) (Fig. 4c). Three cases had a final diagnosis of TB infection rather than virus infection. However, in one case, EBV was detected in both cfDNA and wcDNA, one detected EBV in wcDNA but not in cfDNA, and one detected *Escherichia coli* in wcDNA but not in cfDNA.

## DISCUSSION

Here, we evaluated the clinical utility of mNGS in a patient population with difficult-to-identify pathogens and who were most likely having viral meningitis, encephalitis, or meningomyelitis at the time of testing. We detected a spectrum of DNA virus infections

**a**                                                    **b**

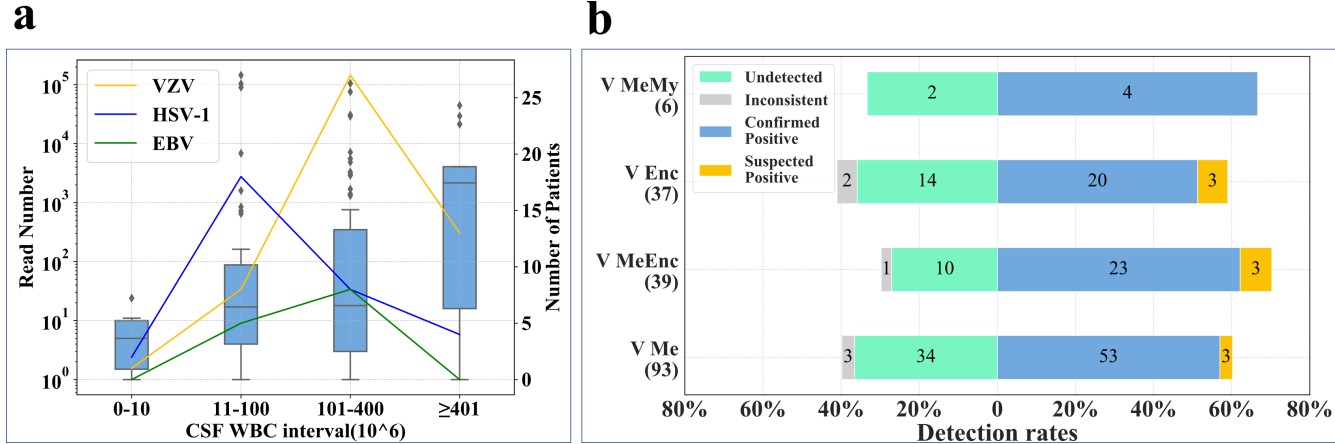

**FIG 3** Correlation between reads and CSF WBC count with the number of patients in each CSF WBC interval (a) and comparison of mNGS detection rates between patients diagnosed with viral encephalitis, meningitis, meningoencephalitis, and meningomyelitis (b). In Panel a, the box chart represents read number and the line chart represents the number of patients. Abbreviations: HSV, herpes simplex virus; VZV, varicella zoster virus; EBV, Epstein–Barr virus; CSF, cerebrospinal fluid; WBC, white blood cell; V, viral; Enc, encephalitis; MeEnc, meningoencephalitis; Me, meningitis; MeMy, meningomyelitis.

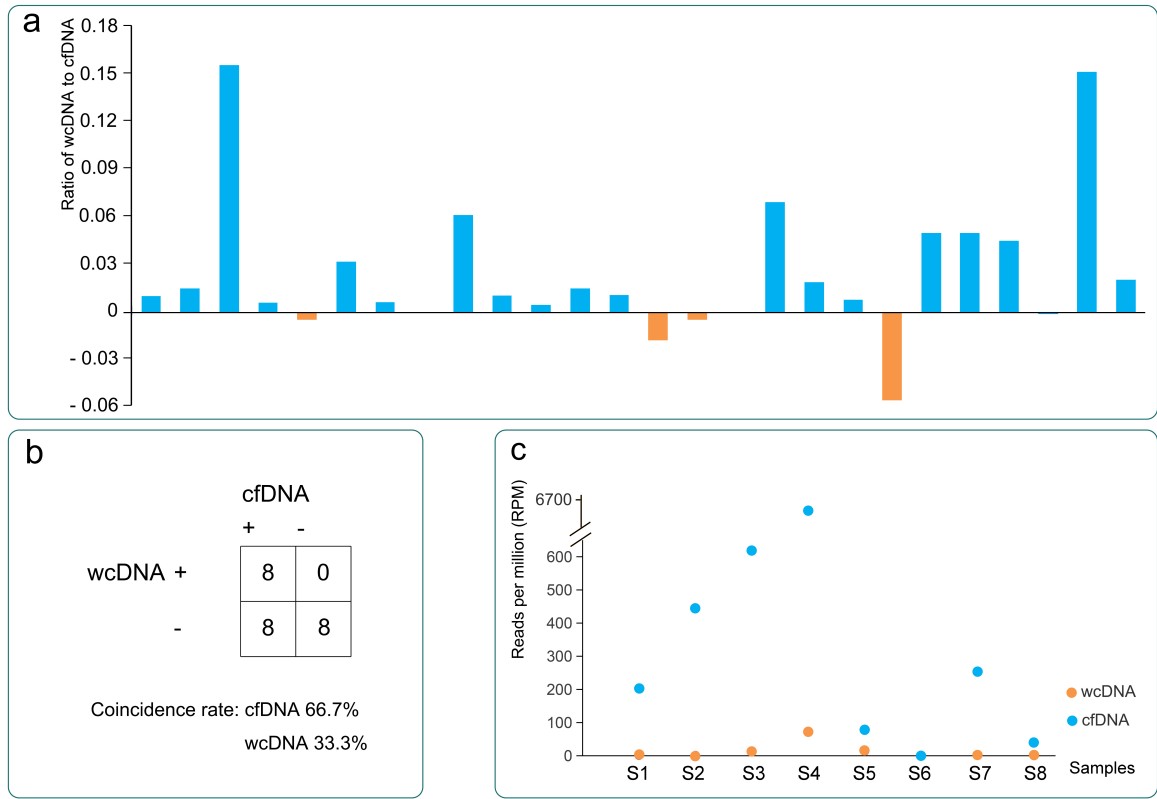

**FIG 4** Comparison of cfDNA and wcDNA results. (a) Comparison of the ratio of microbial DNA between cfDNA mNGS and wcDNA mNGS.The X axis represents the 26 samples tested by both cfDNA and wcDNA. The Y axis represents the ratio of human DNA content in wcDNA and cfDNA data, logarithmically transformed to base 10. When the value in the Y axis is greater than 0, it indicates that the proportion of human DNA detected in wcDNA is higher than that detected in cfDNA in the sample; (b) Virus detection results by mNGS using cfDNA and wcDNA in the 24 cases finally diagnosed with viral CNS infections; (c) The reads per million (RPM) of detected viruses in both cfDNA and wcDNA. The X axis labels S1 to S8 represent the eight samples that tested positive for the same viruses. The Y axis represents the RPM detected by cfDNA and wcDNA in each sample.

in this population, with herpesviruses such as VZV, HSV-1, 2, and EBV particularly common. The extremely specific pathogen identification capability of mNGS is clinically useful, and mNGS of cfDNA performed better for the diagnosis of viruses than mNGS of wcDNA. These findings confirm the important role for clinical mNGS testing in patients with viral CNS infections.

CNS viral infectious diseases depend on the geography and exposure; for example, tick-borne encephalitis virus is common in New England (13), enteroviruses in Finland (25), HSV-1 in France (26) and Australia (27), and JEV in Southeast Asia (28). Our confirmed positive cases were dominated by herpesviruses, including a rarer case of porcine herpesvirus in which no other DNA viruses were identified. The most frequent causative pathogen was VZV (28.0%), followed by HSV-1 (18.3%), consistent with reports of increasing recognition of the role of VZV in CNS pathology (29, 30). It is worth mentioning that although mNGS helped us recognize DNA viruses in this population dominated by herpesviruses, the results of five herpesvirus qPCR assays were not satisfactory, with only 11 out of 50 randomly selected patients (22.9%) testing positive. It proves that the positive detection rate of mNGS in CSF infection is significantly more sensitive than qPCR.

Our results provide clinically relevant guidance to physicians. They show that markedly pleocytic CSF with elevated protein is most likely associated with a VZV etiology. This finding is interesting. The range of WBC counts in patients with VZV was comparable to that of patients with bacterial meningitis and wider than that of

patients with other viral CNS infections, consistent with previous studies (26, 31). CSF protein levels in VZV patients were similar to those in tuberculous meningitis, with 28 (57.1%) VZV patients having CSF protein levels > 1,000 mg/mL and three even exceeding 3,000 mg/mL. Without the support of the mNGS diagnosis, these perplexing data make it clinically highly likely that patients with VZV infection might be misdiagnosed with bacterial or tuberculous meningitis. As VZV CNS infection is thought to be due to the reactivation of a dormant herpes zoster infection, it can be regarded as a secondary immune response. Indeed, Moffat et al. demonstrated a unique tropism of VZV for human T lymphocytes and that release from infected lymphocytes can accelerate the spread of VZV (32). The replication of this otherwise highly cell-associated virus explains the intense leukocytosis seen in the CSF of patients infected with VZV.

While all reported microorganisms met a set positive judgment criteria, in practice, it is difficult to set positive judgment criteria using thresholds alone and without the clinical context. mNGS could not distinguish between invasive infection and colonization or between latent and active infection. For example, EBV is a common virus in humans, and around 95% of people have asymptomatic EBV infection (33), but it was the cause of CNS symptoms (headache, fever, vomiting, neck stiffness, epileptic seizures, and altered mental status) in only 13 patients in our study. EBV can remain dormant in the CNS, and a positive CSF EBV result can occur in latent infection, active infection, EBV-related lymphoproliferative diseases, and EBV infection-associated neuroimmune disorders (34). The number of CSF EBV copies varies between conditions (e.g., higher in CNS EBV lymphoma than EBV encephalitis) and can increase non-specifically with other CNS infections (34, 35). Hence, the significance of CSF EBV depends on the clinical context and should be assessed in conjunction with the patient's presentation, medical history, and immune status, and serial testing and monitoring may be warranted. Three cases of unexpected TTV were also detected. TTV is highly prevalent in the general population and is considered to be an orphan virus (36). An orphan virus refers to a viral species or strain that has been identified and characterized, but its association with any known disease or clinical condition remains unclear or unestablished (37). Nevertheless, recent mNGS-based studies have similarly reported TTV in the CSF of patients with encephalitis/meningitis (38–40). Overall, there is no simple way to determine whether a microorganism is the cause of an individual patient's CNS disease, and a comprehensive epidemiological history, consideration of the clinical presentation, imaging, and CSF and serological findings are required to interpret CSF mNGS reports.

We evaluated the capability of mNGS to detect viruses using cfDNA and wcDNA and found that mNGS using cfDNA showed significant advantages over mNGS using wcDNA. cfDNA reduces interference from human DNA by removing most human cells before extraction. Compared with wcDNA, cfDNA contained a significantly lower proportion of human DNA and higher RPM of pathogens after mNGS. A previous study demonstrated that mNGS of centrifuged supernatants from clinical CSF samples in patients with TBM and cryptococcal meningitis (CM) is a simple and effective method to improve the sensitivity of pathogen detection (17). Results from a recent study evaluating the performance of mNGS using cfDNA in diagnosing CNS infections similarly highlighted the importance of using CSF cfDNA when diagnosing CNS infections, particularly with regard to viruses and MTB (41). Our study confirms the superior viral detection rate in CSF using cfDNA rather than wcDNA. This also explains why the sensitivity of mNGS did not decrease with increasing CSF WBC count but rather increased, in association with elevated mNGS reads (Fig. 3). In addition, seven out of eight patients shared the same detected viruses by both cfDNA and wcDNA, except one with human beta-herpesvirus 7 (HHV-7) detected by wcDNA and HSV-1 by cfDNA, indicating high consistency between the two methods.

Our study has several limitations. First, we did not conduct RNA virus detection, which may have led to false-negative results. RNA viruses, such as enteroviruses, are common causes of encephalitis/meningitis, especially in pediatric patients (4). Thus, DNA/RNA should be simultaneously evaluated to improve virus detection rates in the future.

Additionally, false negatives can also occur through improper specimen collection, transportation, and preservation due to the retrospective nature of the study. Second, there may have been selection bias, as patients had to have a lumbar puncture at some point during admission to be included, and some cases may have been diagnosed and treated in other departments. Third, the small sample size of 50 cases selected for PCR, due to the limited CSF volume and cost constraints, introduces the possibility of bias in the subset of samples used for comparison with the overall data set. Similarly, only CSF wcDNA from 26 randomly selected patients was analyzed by mNGS, which may also introduce bias in the data results. However, we attempted to ensure that the randomly selected samples were representative of the overall sample, and future prospective studies and increased sample size for comparison are needed to evaluate the utility of CSF mNGS in clinical cases of viral encephalitis (meningitis).

## Conclusions

In the context of detecting only DNA viruses by mNGS, it is evident that herpesviruses prominently dominated among patients with CNS viral infections in Hunan Province, China, with VZV being the most common, followed by HSV-1. Patients with VZV infection are characterized by high CSF WBC counts and high read numbers. mNGS, especially using cfDNA, is a useful complementary diagnostic modality to detect viruses causing CNS infection, and further studies with increased sample sizes in wcDNA analysis will be necessary to strengthen the robustness of our findings.

### ACKNOWLEDGMENTS

The authors thank the patients and their families, thank Hugobiotech, Beijing, China, for sharing their mNGS procedure and providing a sequencing data list for this study, and thank all the hospitals that provided CSF samples, including the Third Xiangya Hospital, the Second Xiangya Hospital, the Xiangya Hospital, the People's Hospital of Hunan Province, the First Brain Hospital of Hunan Province, the Fourth Changsha Hospital, the Eighth Changsha Hospital, Xiangtan Central Hospital, the First People's Hospital of Yiyang City, Zhujiang Central Hospital, the First Affiliated Hospital of South China University, and Changsha Central Hospital. The authors also thank the Wisdom Accumulation and Talent Cultivation Project of the Third xiangya hospital of Central South University for funding the reagent costs of the metagenomic sequencing experiment in this study.

This work was supported by the Wisdom Accumulation and Talent Cultivation Project of the Third Xiangya Hospital of Central South University (No.YX202205).

D. L.: Conceptualization, Methodology, and Writing-Reviewing and Editing, Visualization. Y. L.: Data curation, Methodology, Writing-Original draft preparation, Formal analysis, and Visualization.Caiyu Ma: Data curation, Investigation, Methodology, Writing-Original draft preparation, and Validation.Y. Z., Z. L., and X. H.: Supervision, Software, and Data curation. Q. Z., Q. Z., S. Z., H. C., and H. Z.,: Formal analysis and Data curation. Z. S. and R. Z.: Resources.

### AUTHOR AFFILIATIONS

[1]Department of Neurology, The Third Xiangya Hospital of Central South University, Changsha, China

[2]Key laboratory of Microbial Molecular Biology of Hunan Province, Hunan Provincial Center for Disease Control and Prevention, Changsha, China

[3]Department of Scientific Affairs, Hugobiotech Co., Ltd., Beijing, China

[4]Department of Neurology, The Affiliated Changsha Hospital of Xiangya School of Medicine, Central South University, Changsha, China

### AUTHOR ORCIDs

Caiyu Ma http://orcid.org/0000-0002-3682-6096

Ding Liu ⓘ http://orcid.org/0000-0002-9214-7274

## FUNDING

| Funder | Grant(s) | Author(s) |
| --- | --- | --- |
| Wisdom Accumulation and Talent Cultivation Project of the Third xiangya hospital of Central South University | No.YX202205 | Ding Liu |

## ETHICS APPROVAL

The ethics committees of all relevant hospitals reviewed and approved the study protocols (approval no. 22286). All procedures involving human participants were performed in accordance with the ethical standards of the institutional and/or national research committee(s) and the Helsinki Declaration (as revised in 2013). All patients provided written informed consent.

## ADDITIONAL FILES

The following material is available online.

Open Peer Review

**PEER REVIEW HISTORY (review-history.pdf).** An accounting of the reviewer comments and feedback.

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
