## [Reviewer comments · Microbiology Spectrum]

Microbiology Spectrum

Metagenomic next-generation sequencing of cell-free DNA for the identification of viruses causing central nervous system infections

Yuying Lu, Ye Zhang, Zheng Lou, Xiaomin He, Qinghua Zhang, Qingxia Zhang, Shu Zhao, Han Chen, Haixia Zhu, Zhi Song, Ruxu Zhang, Caiyu Ma, and Ding Liu

Corresponding Author(s): Ding Liu, Department of Neurology, The Third Xiangya Hospital of Central South University

Review Timeline:

Submission Date:	June 1, 2023
Editorial Decision:	July 11, 2023
Revision Received:	September 20, 2023
Editorial Decision:	October 16, 2023
Revision Received:	November 21, 2023
Accepted:	November 22, 2023

Editor: Tulip Jhaveri

Reviewer(s): The reviewers have opted to remain anonymous.

Transaction Report:

DOI: <https://doi.org/10.1128/spectrum.02264-23>

July 11, 2023

Dr. Ding Liu
Department of Neurology, The Third Xiangya Hospital of Central South University
No. 138, Tongzipo Road, Yuelu District, Changsha City, Hunan Province, China.
Changsha
China

Re: Spectrum02264-23 (**Metagenomic next-generation sequencing of cell-free DNA for the identification of viruses causing central nervous system infections**)

Dear Dr. Ding Liu:

Thank you for submitting your manuscript to Microbiology Spectrum. This manuscript needs major revisions before being considered for publication. When submitting the revised version of your paper, please provide (1) point-by-point responses to the issues raised by the reviewers as file type "Response to Reviewers," not in your cover letter, and (2) a PDF file that indicates the changes from the original submission (by highlighting or underlining the changes) as file type "Marked Up Manuscript - For Review Only". Please use this link to submit your revised manuscript - we strongly recommend that you submit your paper within the next 60 days or reach out to me. Detailed instructions on submitting your revised paper are below.

Link Not Available

Sincerely,

Tulip Jhaveri

Journals Department
Reviewer comments:

Reviewer #1 (Comments for the Author):

This multicenter retrospective study aimed to assess the effectiveness of metagenomic next-generation sequencing (mNGS) in detecting viral pathogens in patients with suspected viral meningitis. However, upon careful evaluation, it is evident that the study possesses several methodological limitations that warrant extensive revisions to the manuscript. A major concern arises from the fact that most patients included in the study exhibited viral infections caused by VZV, HSV, or EBV, which raises questions regarding whether this represents the optimal use case for mNGS. Notably, these viruses can be readily detected using virus-specific PCR assays and commercially available Filmarray assay, thereby questioning the utility of mNGS in this

particular patient population. Furthermore, the study lacks a suitable microbiologic comparator test, with only a randomly chosen subset of 50 patients having access to such testing. This limitation significantly restricts the conclusions that can be drawn from the study.

Specific points of clarification and revision are required for certain sections of the manuscript.

- lines 144-148 necessitate a thorough explanation of the different categories, including definitions for terms such as "confirmed" and "suspected positive."
- An inconsistency is noted in lines 160-163, where the total number of patients seems to be inaccurate, given that there were 195 patients included but the breakdown indicates a total of 196.
- The retrospective nature of the study also raises queries, as it appears that cerebrospinal fluid (CSF) samples were only tested following patient consent, potentially indicating prospective sample collection. It would be beneficial to elaborate on the process of obtaining retrospective consent and whether it is common for CSF samples to be frozen for future analysis.
- It is important to understand the standard diagnostic strategies for viral meningitis employed at the participating institutions. Are HSV or VZV PCR routinely conducted, and how were the remaining samples handled, given that only 50 were randomly tested?
- The authors categorized the mNGS results into four groups (lines 143-148), necessitating clear definitions for these categories. Specifically, the term "confirmed" requires clarification, including whether it implies microbiologic confirmation and, if so, which specific tests were employed. Furthermore, the second category, where the pathogen could not be excluded, should be elucidated in terms of the differentiating factors.
- Regarding line 164, it is crucial to provide information on the final diagnosis for patients in whom qPCR was conducted but mNGS was not performed.
- In presenting the causes of viral meningitis in the study population, it is important to clarify whether the samples were consecutively collected or if there is a possibility of missing samples due to factors such as patient refusal or weekend testing, as this could introduce potential bias.
- To enhance clarity, it is recommended that the authors focus on presenting the results in separate sections, such as concordant findings, discordant findings, or according to the mNGS categories. This approach would facilitate a more cohesive understanding of the data presented.
- If the main focus of the article is metagenomic sequencing, it is advisable for the authors to emphasize this aspect rather than providing extensive clinical findings. This would ensure that the paper maintains a clear and coherent focus.
- To enhance clarity and organization, it would be beneficial to create a flowchart illustrating the results for the 175 patients, rather than utilizing the current diagram, which may appear cluttered.
- Would recommend to rephrase the sentence in lines 55-57 to improve its clarity and coherence.

Reviewer #2 (Comments for the Author):

Summary: the authors present the diagnostic utility of mNGS on CSF samples from patients presenting with potential viral meningitis, encephalitis or meningoencephalitis. Furthermore, they compare the sensitivity difference between cfDNA and wcDNA. mNGS is a powerful tool to aid in the diagnosis of central nervous system infections without the need of multiple targeted diagnostic assays.

Major issues:

- Introduction line 49, there is mention on the "poor utility" of conventional methods used for diagnosing viral CNS infections. What are the conventional methods referenced here? Since the next sentence talks about mNGS being highly sensitive and rapid to overcome limitations of traditional culture methods.
- Line 124 - Why were only a subset of the available samples for mNGS compared between cfDNA and wcDNA?
- Line 164 - why were only 50 randomly selected samples tested by standard qPCR? mNGS is great but having the potential of HSV causing an infection a fast approach is recommended. Standard of care for CNS infections is qPCR. The discussion is missing comments regarding the cost of mNGS and the length of time needed from specimen collection to result. Furthermore, line 290 claims higher sensitivity of mNGS when compared to qPCR, what is the LOD for the method presented here? All these details are important advantages and disadvantages that the reader need to be aware with any diagnostic method.
- Line 192 is confusing - MTB sequences were found only in 1 case with non-viral infection? Between brackets there is mention of "reads 8" and Figure 2 has a (6) beside MTB. Please clarify these numbers.
- Line 213 - as the authors comment in the discussion, EBV is prevalent in the population and unless the patient is highly immunocompromised and presenting with PTLTD the significance of EBV is not clear. What would the authors suggest as indication and interpretation of the findings in these patients? Line 314 mentions that EBV was found to be the causative pathogen for clinical syndromes of 13 patients - please expand of what these presentations and syndromes were.

- Line 358 - Herpesviruses dominated the spectrum of viruses detected in this study because only DNA was tested, please clarify this in the conclusion statement

Minor issues:

- Throughout the abstract there is spaces missing between words. Please correct.

Reviewer #3 (Comments for the Author):

Thank you for the opportunity to review this manuscript. I commend the investigators on this interesting and important diagnostic-focused work.

This study by Lu et al. is a multicenter retrospective analysis of the utility of metagenomic next generation sequencing (mNGS) on spinal fluid from patients with clinically diagnosed central nervous system (CNS) infection. The investigators compared the results of cell free DNA mNGS (cfDNA), whole cell mNGS (wcDNA) and qPCR. The study period was January, 2018 - July, 2021. Patients were included if they met clinical diagnostic criteria for CNS infection. After diagnostic testing, clinical and laboratory adjudication was performed to characterize results into four categories: 1) positive 2) suspected positive 3) pathogen inconsistent with clinical disease or 4) no causative pathogen detected. 195 patients with suspected viral CNS infection got mNGS and 57.1% of those were considered to have viral infection after testing. 9 patients were considered suspected positive and 34.3% of patients had tests with no responsible pathogen identified. VZV was the most common pathogen, followed by HSV-1 and EBV. White blood cell counts were significantly higher in the CSF of patients with VZV infection. There were also significant difference between pathogen type and ancillary testing (imaging, additional laboratory testing, etc.). Overall, cfDNA detected less human DNA than wcDNA. Additionally, mNGS was more sensitive than qPCR. The investigators notes that without a positive mNGS result for a virus, clinicians may assume that a patient has bacterial meningitis due to high counts of white blood cells when the infection is actually caused by VZV. The authors note that clinical correlation and pre-analytic testing considerations must be adhered to, but that mNGS, especially using cfDNA, is a useful complementary diagnostic modality to detect viruses causing CNS infection.

Major Concerns:

None

Minor Concerns:

1. Line 62, page 5: please capitalize the "M" in "Mycobacterium tuberculosis"
2. Please briefly explain why patients with AIDS were excluded from the study, particularly since they are at increased risk of acquiring CNS infections.
3. Page 8, Lines 142-143: "In cases with differential diagnoses, the multidisciplinary team reached a consensus." I am not clear on what this means. Please explain and consider re-wording this sentence for clarity. Does this mean in the case of a patient with multiple differential diagnosis, the team worked together to agree which of those was most likely?
4. I suggest re-formatting figure 1. Currently, it is unclear which patients were included and excluded because all groups funnel down into the final cohort.
5. Figure 4: For part a, please consider re-naming the Y axis as something other than the variable name used to produce the graph. Is there something that is clearer and easier for the reader to understand? For part C, what is the label for the x-axis?
6. Page 21, line 318: if possible, please provide a short statement explaining orphan virus for the reader.

Staff Comments:

Preparing Revision Guidelines

- Point-by-point responses to the issues raised by the reviewers in a file named "Response to Reviewers," NOT IN YOUR COVER LETTER.
- Upload a compare copy of the manuscript (without figures) as a "Marked-Up Manuscript" file.

- Each figure must be uploaded as a separate file, and any multipanel figures must be assembled into one file.
- Manuscript: A .DOC version of the revised manuscript
- Figures: Editable, high-resolution, individual figure files are required at revision, TIFF or EPS files are preferred

Please return the manuscript within 60 days; if you cannot complete the modification within this time period, please contact me. If you do not wish to modify the manuscript and prefer to submit it to another journal, please notify me of your decision immediately so that the manuscript may be formally withdrawn from consideration by Microbiology Spectrum.

Review manuscript: Metagenomic next-generation sequencing of cell-free DNA for the identification of viruses causing central nervous system infections

Authors: Lu Y, et al.

Summary: the authors present the diagnostic utility of mNGS on CSF samples from patients presenting with potential viral meningitis, encephalitis or meningoencephalitis. Furthermore, they compare the sensitivity difference between cfDNA and wcDNA. mNGS is a powerful tool to aid in the diagnosis of central nervous system infections without the need of multiple targeted diagnostic assays.

Major issues:

- Introduction line 49, there is mention on the “poor utility” of conventional methods used for diagnosing viral CNS infections. What are the conventional methods referenced here? Since the next sentence talks about mNGS being highly sensitive and rapid to overcome limitations of traditional culture methods.
- Line 124 – Why were only a subset of the available samples for mNGS compared between cfDNA and wcDNA?
- Line 164 – why were only 50 randomly selected samples tested by standard qPCR? mNGS is great but having the potential of HSV causing an infection a fast approach is recommended. Standard of care for CNS infections is qPCR. The discussion is missing comments regarding the cost of mNGS and the length of time needed from specimen collection to result. Furthermore, line 290 claims higher sensitivity of mNGS when compared to qPCR, what is the LOD for the method presented here? All these details are important advantages and disadvantages that the reader need to be aware with any diagnostic method.
- Line 192 is confusing – MTB sequences were found only in 1 case with non-viral infection? Between brackets there is mention of “reads 8” and Figure 2 has a (6) beside MTB. Please clarify these numbers.
- Line 213 – as the authors comment in the discussion, EBV is prevalent in the population and unless the patient is highly immunocompromised and presenting with PTLD the significance of EBV is not clear. What would the authors suggest as indication and interpretation of the findings in these patients? Line 314 mentions that EBV was found to be the causative pathogen for clinical syndromes of 13 patients – please expand of what these presentations and syndromes were.
- Line 358 - Herpesviruses dominated the spectrum of viruses detected in this study because only DNA was tested, please clarify this in the conclusion statement

Minor issues:

- Throughout the abstract there is spaces missing between words. Please correct.

Summary of Key Findings (200-250 words)

This study by Lu et al. is a multicenter retrospective analysis of the utility of metagenomic next generation sequencing (mNGS) on spinal fluid from patients with clinically diagnosed central nervous system (CNS) infection. The investigators compared the results of cell free DNA mNGS (cfDNA), whole cell mNGS (wcDNA) and qPCR. The study period was January, 2018 – July, 2021. Patients were included if they met clinical diagnostic criteria for CNS infection. After diagnostic testing, clinical and laboratory adjudication was performed to characterize results into four categories: 1) positive 2) suspected positive 3) pathogen inconsistent with clinical disease or 4) no causative pathogen detected. 195 patients with suspected viral CNS infection got mNGS and 57.1% of those were considered to have viral infection after testing. 9 patients were considered suspected positive and 34.3% of patients had tests with no responsible pathogen identified. VZV was the most common pathogen, followed by HSV-1 and EBV. White blood cell counts were significantly higher in the CSF of patients with VZV infection. There were also significant difference between pathogen type and ancillary testing (imaging, additional laboratory testing, etc.). Overall, cfDNA detected less human DNA than wcDNA. Additionally, mNGS was more sensitive than qPCR. The investigators notes that without a positive mNGS result for a virus, clinicians may assume that a patient has bacterial meningitis due to high counts of white blood cells when the infection is actually caused by VZV. The authors note that clinical correlation and pre-analytic testing considerations must be adhered to, but that mNGS, especially using cfDNA, is a useful complementary diagnostic modality to detect viruses causing CNS infection.

Major Concerns (at most 5-6):

None

Minor Concerns (at most 5-20 in bullet points):

1. Line 62, page 5: please capitalize the “M” in “Mycobacterium tuberculosis”
2. Please briefly explain why patients with AIDS were excluded from the study, particularly since they are at increased risk of acquiring CNS infections.
3. Page 8, Lines 142-143: “In cases with differential diagnoses, the multidisciplinary team reached a consensus.” I am not clear on what this means. Please explain and consider re-wording this sentence for clarity. Does this mean in the case of a patient with multiple differential diagnosis, the team worked together to agree which of those was most likely?
4. I suggest re-formatting figure 1. Currently, it is unclear which patients were included and excluded because all groups funnel down into the final cohort.
5. Figure 4: For part a, please consider re-naming the Y axis as something other than the variable name used to produce the graph. Is there something that is clearer and easier for the reader to understand? For part C, what is the label for the x-axis?
6. Page 21, line 318: if possible, please provide a short statement explaining orphan virus for the reader.

Reviewer #1

This multicenter retrospective study aimed to assess the effectiveness of metagenomic next-generation sequencing (mNGS) in detecting viral pathogens in patients with suspected viral meningitis. However, upon careful evaluation, it is evident that the study possesses several methodological limitations that warrant extensive revisions to the manuscript. A major concern arises from the fact that most patients included in the study exhibited viral infections caused by VZV, HSV, or EBV, which raises questions regarding whether this represents the optimal use case for mNGS. Notably, these viruses can be readily detected using virus-specific PCR assays and commercially available Filmarray assay, thereby questioning the utility of mNGS in this particular patient population. Furthermore, the study lacks a suitable microbiologic comparator test, with only a randomly chosen subset of 50 patients having access to such testing. This limitation significantly restricts the conclusions that can be drawn from the study.

Response: We appreciate the valuable feedback from the reviewers. We acknowledge the concern raised about the predominance of VZV, HSV, or EBV infections in our patient cohort, these viruses can indeed be detected using virus-specific PCR assays and commercially available Filmarray assays. However, in our clinical experience, few viral detection methods are routinely used in Chinese hospitals. While PCR is capable of detecting common pathogens, its positivity rate in CSF is not ideal, especially when dealing with low levels of viral genetic material. This can result in false-negative results, which can be critical in clinical decision-making. Additionally, the virological basis of viral meningitis cases can be complex, and qPCR alone may not provide a comprehensive characterization. Therefore, we argue that mNGS offers advantages in sensitivity and the ability to identify a broader range of pathogens, making it a valuable tool in patients with suspected viral encephalitis.

We also acknowledge the limitation of not having a suitable microbiologic comparator test for all patients in our study. As the reviewers noted, we did include a subset of 50 patients who underwent microbiological comparison testing. We understand that this sample size is limited, and the availability of suitable specimens was constrained by various factors, including nucleic acid degradation over time, the practical capacity of cerebrospinal fluid specimens, potential negative PCR results

within the first 72 hours of disease onset, prior antiviral treatment, budget constraints, and other logistical challenges.

In response to these limitations, we have already mentioned them in the Discussion section of our manuscript, emphasizing the need for further studies with larger sample sizes and more comprehensive comparisons. We understand the importance of robust validation and intend to explore additional avenues for expanding our analyses in future research efforts.

Specific points of clarification and revision are required for certain sections of the manuscript.

- lines 144-148 necessitate a thorough explanation of the different categories, including definitions for terms such as "confirmed" and "suspected positive."

Response: Thank you for your valuable suggestions. We have made revisions to the revised manuscript based on your recommendations. The updated content is as follows: We categorized the mNGS results into four groups: (i) Confirmed Positive: where mNGS results definitively proved the presence of a neuroinvasive pathogen, consistent with the clinical diagnosis and confirmed by traditional microbiological detection methods (PCR, antibody detection, or culture) , or consistent with diagnosis of clinical experience and no other CNS diseases were evident during follow-up, (ii) Suspected Positive: where mNGS detected pathogens potentially related to the disease, which could be latent infection, viral carrier status, active infections, or a potential new, rare, or unexpected pathogen, requiring further confirmation through microbiological methods. (iii) Inconsistent with Clinical Diagnosis: where the mNGS results presented a pathogen that did not align with the clinical diagnosis or expected disease profile. (iv) Causative Pathogens Not Detected: where the mNGS results didn't identify any pathogen responsible for the disease. See page 9, lines 164-174 of the manuscript for details.

- An inconsistency is noted in lines 160-163, where the total number of patients seems to be inaccurate, given that there were 195 patients included but the breakdown indicates a total of 196.

Response: We sincerely apologize for this error. We stated in the original manuscript that there were eight non-infectious CNS cases when it was actually seven non-infectious CNS cases, thus creating an inconsistency in the numbers. We have corrected the number of non-infectious CNS cases in page 10, line 191 of the revised

manuscript to seven. Your feedback has been immensely helpful in ensuring the accuracy of our work.

- The retrospective nature of the study also raises queries, as it appears that cerebrospinal fluid (CSF) samples were only tested following patient consent, potentially indicating prospective sample collection. It would be beneficial to elaborate on the process of obtaining retrospective consent and whether it is common for CSF samples to be frozen for future analysis.

Response: Your suggestion is very necessary, the following is our reply to this review comment: While our study is retrospective, CSF samples were collected prospectively as part of routine clinical procedures. The samples were not collected specifically for this study. We added clarifications in the Methods section of the revised manuscript (see page 6, lines 100-103 of the revised manuscript for details). It is common practice in many medical facilities to freeze and store excess CSF samples for potential future analysis. These samples can be valuable for research purposes, as they allow for investigations beyond the initial clinical diagnosis. To address ethical issues associated with possible future retrospective studies using patient data and samples, we obtained patient consent at the time of sample collection.

- It is important to understand the standard diagnostic strategies for viral meningitis employed at the participating institutions. Are HSV or VZV PCR routinely conducted, and how were the remaining samples handled, given that only 50 were randomly tested?

Response: Thank you for the inquiry from the esteemed reviewer. We acknowledge the importance of understanding standard diagnostic strategies for viral encephalitis (meningitis). The diagnostic strategy for viral encephalitis (meningitis) at our participating institution is confirmed based on clinical presentation, laboratory and ancillary test results, treatment response, and the exclusion of other CNS diseases, as outlined in our patient inclusion and exclusion criteria. None of the 195 patients finally included had etiological diagnostic tests performed before mNGS. We explain as follows: Currently, few viral detection methods are routinely used in Chinese hospitals. While PCR is capable of detecting common pathogens, its positivity rate in CSF is not ideal, and it necessitates a priori pathogen designation, making it challenging to meet clinical diagnostic demands. As demonstrated by the results from our randomly selected cohort of 50 patients, only 11 cases (22.9%) returned positive

PCR results. The use of PCR in clinical practice for CNS infections is so low that most hospitals no longer use it. Therefore, HSV or VZV PCR testing is not routinely performed. However, mNGS has brought new perspectives to CNS infection investigations and is expected to solve the etiological deficiencies in the field of CNS infections. We apologize for not clearly stating our diagnostic strategy for the patients included in the manuscript. We have provided clarification in the revised version to ensure that readers have a clear understanding of the diagnostic strategy for the patients we included. see page 7, lines 122-125 of the revised manuscript for details.

- The authors categorized the mNGS results into four groups (lines 143-148), necessitating clear definitions for these categories. Specifically, the term "confirmed" requires clarification, including whether it implies microbiologic confirmation and, if so, which specific tests were employed. Furthermore, the second category, where the pathogen could not be excluded, should be elucidated in terms of the differentiating factors.

Response: We greatly appreciate your identification of issues in our article, and we have made revisions based on your suggestions. Specifically, the term "confirmed" refers to the pathogen detected by mNGS being confirmed to be consistent with diagnosis of clinical experience and confirmed by qPCR or confirmed to be consistent with diagnosis of clinical experience with no other CNS diseases during follow-up. The second category, "suspected positive," denotes pathogens detected by mNGS that may be related to the disease, and the results need to be combined with the clinical situation and repeated tests to determine whether they are latent infection, virus carrier status or active infection; or the detection of a potential novel, rare or unexpected pathogen, further microbiological methods are needed to confirm and determine its significance. We have made corresponding changes in the revised manuscript, see page 9, lines 164-171 of the revised manuscript.

- Regarding line 164, it is crucial to provide information on the final diagnosis for patients in whom qPCR was conducted but mNGS was not performed.

Response: We apologize for any confusion that may have arisen from the wording in line 164 of the original manuscript. Indeed, we would like to clarify that all enrolled patients were subject to mNGS, only 50 cases underwent qPCR analysis. We are grateful for the opportunity to provide this clarification and rewrite it in the revised manuscript to avoid this confusion.

- In presenting the causes of viral meningitis in the study population, it is important to clarify whether the samples were consecutively collected or if there is a possibility of missing samples due to factors such as patient refusal or weekend testing, as this could introduce potential bias.

Response: We acknowledge the importance of addressing potential biases in the study population when presenting the causes of viral meningitis. In our study, the samples were consecutively collected from eligible patients who met the inclusion criteria. It is essential to note that there is a possibility of missing samples due to factors such as patient refusal or weekend testing. We have taken measures to minimize this potential bias by thoroughly documenting the reasons for any missing samples and evaluating whether there are any significant differences between the included and excluded cases. We will clarify the continuity of sample collection and ensure transparency in the Methods section of the revised manuscript, see page 6, lines 97-99 for details.

- To enhance clarity, it is recommended that the authors focus on presenting the results in separate sections, such as concordant findings, discordant findings, or according to the mNGS categories. This approach would facilitate a more cohesive understanding of the data presented.

Response: We appreciate the reviewers' valuable comments and believe that revising the manuscript in accordance with these comments has significantly enhance the overall quality of our manuscript. We have restructured the presentation of results, organizing them into separate sections based on mNGS result category. See page 11, lines 197-211 of the revised manuscript for details.

- If the main focus of the article is metagenomic sequencing, it is advisable for the authors to emphasize this aspect rather than providing extensive clinical findings. This would ensure that the paper maintains a clear and coherent focus.

Response: Thank you for your valuable feedback. In response to your suggestion, we have revised the manuscript to prioritize the metagenomic sequencing aspect while ensuring that the clinical findings are appropriately presented to support and contextualize our research. This will help improve the clarity and coherence of our paper, aligning it more closely with the primary focus on mNGS.

- *To enhance clarity and organization, it would be beneficial to create a flowchart illustrating the results for the 175 patients, rather than utilizing the current diagram, which may appear cluttered.*

Response: Thank you for your valuable input regarding the clarity and organization of our manuscript. In response to your suggestion, we have created a clear and concise flowchart that effectively summarizes the key findings for all 175 patients in a more organized manner. See Figure2 (page 12) of the revised manuscript for details.

- *Would recommend to rephrase the sentence in lines 55-57 to improve its clarity and coherence.*

Response: Thanks for your suggestion, we have made changes in the revised manuscript in lines 70-73 as follows: The effectiveness of pathogen detection through CSF mNGS is notably affected by the presence of human DNA contamination, as highlighted in prior research. White blood cells (WBCs), commonly found in CSF from patients with CNS infections, serve as a primary source of host-related impurities.

Reviewer #2

Summary: the authors present the diagnostic utility of mNGS on CSF samples from patients presenting with potential viral meningitis, encephalitis or meningoencephalitis. Furthermore, they compare the sensitivity difference between cfDNA and wcDNA. mNGS is a powerful tool to aid in the diagnosis of central nervous system infections without the need of multiple targeted diagnostic assays.

Major issues:

- *Introduction line 49, there is mention on the "poor utility" of conventional methods used for diagnosing viral CNS infections. What are the conventional methods referenced here? Since the next sentence talks about mNGS being highly sensitive and rapid to overcome limitations of traditional culture methods.*

Response: Thank you for your valuable suggestions. Our reply is as follows: The term "conventional methods" refers to the traditional diagnostic techniques commonly used for diagnosing viral CNS infections. These methods may include PCR tests, antibody detection (enzyme immunoassays), and viral culture. The mention of "poor utility" of conventional methods is based on their limitations in terms of sensitivity,

specificity, and turnaround time. These conventional methods often face challenges in detecting low levels of viral load, leading to false-negative results in some cases. Additionally, the time required for sample processing and obtaining results may delay the diagnosis, which can be critical in cases of CNS infections where prompt treatment is essential. We have modified this sentence accordingly in lines 60-64 in the revised manuscript as follows: Furthermore, the viral pathogen spectrum detected in CNS infections is typically complex and diverse, with potentially low viral loads. Conventional methods such as PCR, antibody detection (enzyme immunoassays), and viral culture often exhibit poor utility in such cases.

- *Line 124 - Why were only a subset of the available samples for mNGS compared between cfDNA and wcDNA?*

Response: We appreciate the reviewer's question regarding the subset of samples used for mNGS comparison between cfDNA and wcDNA. The decision to compare a subset of available samples for mNGS analysis between cfDNA and wcDNA was influenced by several factors, primarily centered around practicality and resource limitations. Conducting mNGS on the entire set of available samples would have required a considerable amount of both financial and time resources. By opting to perform the comparison on a subset of samples, we aimed to achieve a meaningful evaluation of the performance of cfDNA and wcDNA while efficiently utilizing our available resources.

We acknowledge that this approach does introduce the potential for selection bias within the subset of samples chosen for comparison. However, we took measures to ensure that the subset was representative of the broader sample population, and we have highlighted this limitation in the discussion of our findings (page 21, lines 372-377).

In light of the reviewer's query, we recognize the importance of addressing this limitation and will consider expanding our analysis to include a more comprehensive comparison in future studies, thus enhancing the robustness and generalizability of our results.

- *Line 164 - why were only 50 randomly selected samples tested by standard qPCR? mNGS is great but having the potential of HSV causing an infection a fast approach is recommended. Standard of care for CNS infections is qPCR. The discussion is missing comments regarding the cost of mNGS and the length of time needed from*

specimen collection to result. Furthermore, line 290 claims higher sensitivity of mNGS when compared to qPCR, what is the LOD for the method presented here? All these details are important advantages and disadvantages that the reader need to be aware with any diagnostic method.

Response: Thank you very much for your comments, the details you have raised are important. We selected only 50 specimens for qPCR testing due to several factors. Firstly, the extended freezing time of some CSF samples can lead to nucleic acid degradation, affecting the reliability of qPCR results. In our experiment, qPCR results for the five most common herpesviruses yielded unsatisfactory outcomes, with only 11 out of 50 randomly selected patients testing positive. Moreover, the limited CSF volume, the potential for negative PCR results within 72 hours before the onset of disease, and cost constraints were among the considerations that influenced our decision to analyze only 50 specimens via qPCR. We acknowledge that this limitation is a significant drawback of our study, and we have discussed it in our revised manuscript (pages 20-21, lines 369-372).

We appreciate your suggestion regarding the inclusion of comments about the cost of mNGS and the turnaround time from specimen collection to results. However, we did not address these aspects in our study for the following reasons: Our study was retrospective, and many of the specimens were frozen. Consequently, the time taken from specimen collection to results varied significantly and would not provide informative data for our analysis. While we acknowledge that cost is an essential factor in diagnostic methods, our study did not involve a direct cost comparison between mNGS and qPCR. Detailed cost analysis would require a separate study with a focus on cost-effectiveness.

We apologize for not providing the Limit of Detection (LOD) for the mNGS method presented in our study. Unfortunately, we did not determine the LOD in this specific research. As discussed elsewhere, the broad-range nature of the technique makes it difficult to determine the sensitivity or LOD for every possible organism(1). Future studies may aim to address this important parameter to provide a comprehensive evaluation of the diagnostic method.

We appreciate your feedback and will consider these aspects for future research and discussions.

- Line 192 is confusing - MTB sequences were found only in 1 case with non-viral

infection? Between brackets there is mention of "reads 8" and Figure 2 has a (6) beside MTB. Please clarify these numbers.

Response: Apologies for any confusion. Allow me to provide a clearer explanation now: The phrase " MTB sequences (reads 8) were detected in only one of eight diagnoses of CNS tuberculous infection" means that in 8 patients with tuberculous meningitis, MTB sequence was detected in only 1 patient, with 8 sequence reads. There is a (6) beside MTB in the original Figure 2, indicating that MTB sequence detected in 6 patients with viral meningitis/encephalitis, the numbers in the () indicate is the number of cases. The mNGS results of these 6 patients were inconsistent with the diagnosis. We have redone Figure 2 and also reworded to avoid this confusion: 8 cases were diagnosed as CNS tuberculosis infection, with only 1 case detecting MTB sequence through mNGS. See page 11, lines 212-213 and Figure 2 (page 12) of the revised manuscript for details.

- Line 213 - as the authors comment in the discussion, EBV is prevalent in the population and unless the patient is highly immunocompromised and presenting with PTLD the significance of EBV is not clear. What would the authors suggest as indication and interpretation of the findings in these patients? Line 314 mentions that EBV was found to be the causative pathogen for clinical syndromes of 13 patients - please expand of what these presentations and syndromes were.

Response: Thanks to the reviewer for paying attention to this issue. EBV is indeed prevalent in the general population, and its mere presence does not necessarily imply clinical significance in all cases. The EBV can establish latency within the CNS, and a positive CSF EBV nucleic acid result can be observed in latent infection, active infection, EBV-related lymphoproliferative diseases, and EBV infection-associated neuroimmune disorders(2). EBV may also be reactivated secondary to other CNS infections, and an increased number of EBV copies may be unspecific(3). There is no consensus on the minimal number of copies in the CSF required for a diagnosis of EBV encephalitis(meningitis). Therefore, the indications and interpretations of EBV findings in these patients should be approached with a nuanced perspective. We would like to emphasize that the significance of CSF EBV should be assessed in relation to the patient's clinical presentation, medical history, and immune status. If a patient presents with typical symptoms of viral encephalitis or meningitis (e.g., headache, fever, vomiting, neck stiffness, epileptic seizures, and altered mental status),

the presence of CSF EBV may be investigated as a causative agent. In cases where EBV is detected, serial testing and monitoring may be warranted. A single positive result may not provide a complete picture of the infection's status. Repeated testing over time can help determine whether the EBV viral load is increasing, decreasing, or remaining stable, which can provide valuable information regarding the course of the infection. In individuals with intact immune systems, EBV infections are often well-controlled and may not cause severe pathology. However, in cases of immunosuppression, such as in transplant recipients or individuals with compromised immune function, EBV reactivation can lead to more severe clinical outcomes. We will add relevant content to the Discussion section of the revised manuscript as appropriate. See pages 18-19, lines 323-333 or details.

We appreciate the reviewer's inquiry regarding the clinical syndromes of the 13 patients in whom EBV was identified as the causative pathogen. As with other forms of viral encephalitis(meningitis), clinical presentations and syndromes of EBV encephalitis(meningitis), include headache, fever, vomiting, neck stiffness, epileptic seizures, and altered mental status. As in the 13 EBV-positive patients listed in Table 1 in our manuscript. We also have emphasized this point in the Discussion section of the revised manuscript.

- Line 358 - Herpesviruses dominated the spectrum of viruses detected in this study because only DNA was tested, please clarify this in the conclusion statement

Minor issues:

Response: Your suggestion has been very useful to our article, and we have made the following changes to page 21, lines 379-381 of the revised manuscript: In the context of detecting only DNA viruses within this study, it is evident that herpesviruses prominently dominated among patients with CNS viral infections in Hunan Province, China, with VZV the most common, followed by HSV-1.

- Throughout the abstract there is spaces missing between words. Please correct.

Response: We are very sorry for this undeserved error. Modifications have been made in the revised manuscript.

Reviewer #3

Thank you for the opportunity to review this manuscript. I commend the investigators on this interesting and important diagnostic-focused work.

This study by Lu et al. is a multicenter retrospective analysis of the utility of metagenomic next generation sequencing (mNGS) on spinal fluid from patients with clinically diagnosed central nervous system (CNS) infection. The investigators compared the results of cell free DNA mNGS (cfDNA), whole cell mNGS (wcDNA) and qPCR. The study period was January, 2018 - July, 2021. Patients were included if they met clinical diagnostic criteria for CNS infection. After diagnostic testing, clinical and laboratory adjudication was performed to characterize results into four categories: 1) positive 2) suspected positive 3) pathogen inconsistent with clinical disease or 4) no causative pathogen detected. 195 patients with suspected viral CNS infection got mNGS and 57.1% of those were considered to have viral infection after testing. 9 patients were considered suspected positive and 34.3% of patients had tests with no responsible pathogen identified. VZV was the most common pathogen, followed by HSV-1 and EBV. White blood cell counts were significantly higher in the CSF of patients with VZV infection. There were also significant difference between pathogen type and ancillary testing (imaging, additional laboratory testing, etc.). Overall, cfDNA detected less human DNA than wcDNA. Additionally, mNGS was more sensitive than qPCR. The investigators notes that without a positive mNGS result for a virus, clinicians may assume that a patient has bacterial meningitis due to high counts of white blood cells when the infection is actually caused by VZV. The authors note that clinical correlation and pre-analytic testing considerations must be adhered to, but that mNGS,

especially using cfDNA, is a useful complementary diagnostic modality to detect viruses causing CNS infection.

Major Concerns:

None

Minor Concerns:

1. Line 62, page 5: please capitalize the "M" in "Mycobacterium tuberculosis"

Response: Thank you for pointing out our error, we have corrected it in the revised manuscript.

2. Please briefly explain why patients with AIDS were excluded from the study, particularly since they are at increased risk of acquiring CNS infections.

Response: We appreciate the opportunity to clarify the exclusion of patients with AIDS from our study. While it is true that individuals with AIDS are at an increased risk of acquiring CNS infections, we decided to exclude them from our study due to several reasons: 1. Our study aims to investigate the causes of viral meningitis in a general population, if including patients with AIDS could introduce confounding factors related to their underlying condition and treatments. By focusing on a specific population without AIDS, we can better isolate and analyze the factors contributing to viral meningitis in this group. 2. CNS infections in patients with AIDS are often caused by opportunistic pathogens and may require different diagnostic approaches and treatments compared to viral meningitis in immunocompetent individuals. The inclusion of AIDS patients could lead to significant heterogeneity in the study population, making it challenging to draw meaningful conclusions. 3. Given the relatively low prevalence of AIDS in our study population, the inclusion of such cases might result in a disproportionately small subgroup, limiting the statistical power to detect significant associations with other variables of interest. 4. Research involving patients with AIDS requires additional ethical considerations, as these individuals may be more vulnerable and require specialized care. By excluding them from our study, we can avoid potential ethical complexities and ensure a more straightforward analysis of the data.

3. *Page 8, Lines 142-143: "In cases with differential diagnoses, the multidisciplinary team reached a consensus." I am not clear on what this means. Please explain and consider re-wording this sentence for clarity. Does this mean in the case of a patient with multiple differential diagnosis, the team worked together to agree which of those was most likely?*

Response: Your understanding is correct. We apologize for not making the meaning clear. We provide a clearer reworded version of this sentence in the revised manuscript: "When faced with various potential diagnoses, the multidisciplinary team collaboratively arrived at a consensus to determine the most probable diagnosis." (page 9, lines 162-163)

4. *I suggest re-formatting figure 1. Currently, it is unclear which patients were included and excluded because all groups funnel down into the final cohort.*

Response: Thanks for pointing out our problem. We have made changes to Figure 1(page10) in the revised manuscript to ensure that the flow diagram is clear enough.

5. *Figure 4: For part a, please consider re-naming the Y axis as something other than the variable name used to produce the graph. Is there something that is clearer and easier for the reader to understand? For part C, what is the label for the x-axis?*

Response: Thank you for your advice. The Figure 4 (page16) has been revised accordingly. In part a, the Y axis represents the ratio of human DNA content in wcDNA and cfDNA data, logarithmically transformed to base 10. When the value is greater than 0, it indicates that the proportion of human DNA detected in wcDNA is higher than cfDNA in the sample. For part C, the x-axis labels S1 to S8 represent the eight samples that were tested positive for the same viruses. We have included this additional information in the figure note. See revised manuscript for details.

6. *Page 21, line 318: if possible, please provide a short statement explaining orphan virus for the reader.*

Response: Thank you for your valuable suggestions, we have made corresponding changes in the revised manuscript in Page 19, Lines 335-337. The specific content is: An orphan virus refers to a viral species or strain that has been identified and characterized, but its association with any known disease or clinical condition remains unclear or unestablished(4).

REFERENCES

1. Brown JR, Bharucha T, Breuer J. 2018. Encephalitis diagnosis using metagenomics: application of next generation sequencing for undiagnosed cases. *J Infect* 76:225-240.
2. Weinberg A, Li S, Palmer M, Tyler KL. 2002. Quantitative CSF PCR in Epstein-Barr virus infections of the central nervous system. *Ann Neurol* 52:543-8.
3. Andersen O, Ernberg I, Hedstrom AK. 2023. Treatment Options for Epstein-Barr Virus-Related Disorders of the Central Nervous System. *Infect Drug Resist* 16:4599-4620.

4. Mortimer PP. 2013. Orphan viruses, orphan diseases: still the raw material for virus discovery. *Rev Med Virol* 23:337-9.

October 16, 2023

Dr. Ding Liu
Department of Neurology, The Third Xiangya Hospital of Central South University
No. 138, Tongzipo Road, Yuelu District, Changsha City, Hunan Province, China.
Changsha
China

Re: Spectrum02264-23R1 (**Metagenomic next-generation sequencing of cell-free DNA for the identification of viruses causing central nervous system infections**)

Dear Dr. Ding Liu:

Thank you for submitting your manuscript to Microbiology Spectrum. This manuscript needs further revisions before it can be considered for publication. When submitting the revised version of your paper, please provide (1) point-by-point responses to the issues raised by the reviewers as file type "Response to Reviewers," not in your cover letter, and (2) a PDF file that indicates the changes from the original submission (by highlighting or underlining the changes) as file type "Marked Up Manuscript - For Review Only". Please use this link to submit your revised manuscript - we strongly recommend that you submit your paper within the next 60 days or reach out to me. Detailed instructions on submitting your revised paper are below.

Link Not Available

Sincerely,

Tulip Jhaveri

Journals Department
Reviewer comments:

Reviewer #2 (Comments for the Author):

Most of the issues were addressed in these revisions. Some outstanding concerns remain:

Major issues:

- In the importance they compare mNGS to conventional diagnostic procedures as "time-consuming" - qPCR is not more time consuming than mNGS.
- If this is a retrospective study - what is the gold standard? How was the diagnosis made on these patients?
- The data is quite comprehensive for the panel were cfDNA mNGS was performed on 175 samples. To draw major conclusions with only 26 samples performed on wcDNA.

- What about RNA viruses? This methodology did not allow their recovery. Enterovirus is a common cause of viral meningitis. This should at minimum be addressed in the limitations of the study.
- Line 70 - unclear what the authors mean as traditional culture for the diagnostic of viruses? Is mNGS more sensitive and rapid for pathogen identification than qPCR (that should be the standard of care for the diagnosis of these viruses)? - more relevant literature should be cited here.

Minor issues:

- Throughout the abstract there is spaces missing between words. Please correct - there is a few outstanding

Staff Comments:

Preparing Revision Guidelines

Please return the manuscript within 60 days; if you cannot complete the modification within this time period, please contact me. If you do not wish to modify the manuscript and prefer to submit it to another journal, please notify me of your decision immediately so that the manuscript may be formally withdrawn from consideration by Microbiology Spectrum.

Review manuscript: Metagenomic next-generation sequencing of cell-free DNA for the identification of viruses causing central nervous system infections

Authors: Lu Y, et al.

Summary: the authors present the diagnostic utility of mNGS on CSF samples from patients presenting with potential viral meningitis, encephalitis or meningoencephalitis. Furthermore, they compare the sensitivity difference between cfDNA and wcDNA. mNGS is a powerful tool to aid in the diagnosis of central nervous system infections without the need of multiple targeted diagnostic assays.

Most of the issues were addressed in these revisions. Some outstanding concerns remain:

Major issues:

- In the importance they compare mNGS to conventional diagnostic procedures as “time-consuming” – qPCR is not more time consuming than mNGS.
- If this is a retrospective study – what is the gold standard? How was the diagnosis made on these patients?
- The data is quite comprehensive for the panel were cfDNA mNGS was performed on 175 samples. To draw major conclusions with only 26 samples performed on wcDNA.
- What about RNA viruses? This methodology did not allow their recovery. Enterovirus is a common cause of viral meningitis. This should at minimum be addressed in the limitations of the study.
- Line 70 – unclear what the authors mean as traditional culture for the diagnostic of viruses? Is mNGS more sensitive and rapid for pathogen identification than qPCR (that should be the standard of care for the diagnosis of these viruses)?? – more relevant literature should be cited here.

Minor issues:

- Throughout the abstract there is spaces missing between words. Please correct – there is a few outstanding

Major issues:

- In the importance they compare mNGS to conventional diagnostic procedures as "time consuming" – qPCR is not more time consuming than mNGS.

Response: Thank you for your valuable feedback on our manuscript. We appreciate your suggestions and have carefully considered them.

Regarding the comparison of mNGS to conventional diagnostic procedures, we apologize for any confusion caused by our original manuscript. We understand that qPCR is not inherently more time-consuming than mNGS. The time consumption we mentioned was specifically referring to virus culture, which is not routinely used in clinical laboratories. qPCR or antibody testing are the primary conventional diagnostic methods used. Therefore, we agree with your point that conventional diagnostic methods do not have the issue of being more time-consuming than mNGS.

To address this, we have removed the description of virus culture from the original manuscript to avoid any misinterpretation. We have revised the relevant sections accordingly to ensure clarity and accuracy.

- If this is a retrospective study – what is the gold standard? How was the diagnosis made on these patients?

Response: Thank you for your valuable feedback. In our study, the enrolled participants comprised individuals diagnosed with suspected CNS viral infection who had not undergone pathogen confirmation testing. Therefore, the diagnosis was based on a combination of clinical presentation, laboratory results, imaging findings, and the exclusion of other potential causes. The clinical diagnostic criteria of viral encephalitis, meningitis, meningoencephalitis, or meningomyelitis necessitate simultaneous fulfillment of the following three conditions: a) Major criteria: Altered mental status, encompassing a decline in the level of consciousness, lethargy, or abnormal mental behavior persisting for ≥ 24 hours; or the occurrence of new-onset seizures. b) Minor criteria: Fever (body temperature ≥ 38 °C) either preceding or manifesting within 72 hours after onset, or newly emerging focal neurological manifestations, or

CSF pleocytosis (>5 WBC/ μ L), or electroencephalographic (EEG) findings indicating encephalitis, or abnormal neuroimaging suggesting encephalitis, meningitis, meningoencephalitis, or meningomyelitis. c) The reasonable exclusion of alternative etiologies. We followed the established diagnostic criteria and guidelines recommended by the relevant medical societies and organizations (1). These diagnostic criteria have been widely used and accepted in clinical practice. We acknowledge that the retrospective nature of our study may have limitations in terms of diagnostic accuracy. However, we made efforts to ensure the reliability of the diagnoses by involving experienced clinicians and specialists in the evaluation process. Additionally, we conducted a thorough review of medical records and utilized standardized diagnostic criteria to minimize potential biases. We have clearly stated the diagnostic criteria for our study participants in the Methods section to ensure clarity of our diagnostic procedures to the readers.

We hope this clarifies the diagnostic approach used in our study. If you have any further questions or concerns, please feel free to let us know.

- The data is quite comprehensive for the panel were cfDNA mNGS was performed on 175 samples. To draw major conclusions with only 26 samples performed on wcDNA.

Response: Thank you for providing insightful comments on our study. We value your feedback and have taken your suggestions into careful consideration.

Regarding the issue you raised about the sample size, we acknowledge that the number of samples analyzed using wcDNA in our study is relatively small (26 samples). We agree that this sample size may limit the reliability and accuracy of our conclusions, and we have discussed this limitation in the Discussion section. Although the sample size for wcDNA analysis is limited, the differences observed compared to the analysis of cell-free DNA (cfDNA) are evident.

We understand the importance of a larger sample size for robust statistical analysis and drawing more definitive conclusions. In the

Conclusions section, we have provided a nuanced and balanced perspective, ensuring readers are well-informed about the inherent limitations in our findings, see page 19, lines 355-357 of the revised manuscript.

. We hope this revised response addresses your concerns appropriately. If you have any further questions or suggestions, please feel free to let us know.

- What about RNA viruses? This methodology did not allow their recovery. Enterovirus is a common cause of viral meningitis. This should at minimum be addressed in the limitations of the study.

Response: Thank you for your feedback and suggestions. We appreciate your input and hope to address your concerns. Regarding RNA viruses, as mentioned in the discussion section of our manuscript, one limitation of our study is the lack of mNGS analysis for RNA viruses. We recognize that RNA viruses, such as enteroviruses, are common causes of viral meningitis. As previously described in the literature, enteroviruses are important pathogens causing viral meningitis in pediatric patients in our country (2). We have added a statement in the limitations section of our manuscript to address the issue of not detecting RNA viruses and its potential impact on our results.

- Line 70 – unclear what the authors mean as traditional culture for the diagnostic of viruses? Is mNGS more sensitive and rapid for pathogen identification than qPCR (that should be the standard of care for the diagnosis of these viruses)?? – more relevant literature should be cited here.

Response: We appreciate the reviewer for pointing out these issues. In response to the first question, we apologize for the confusion caused by the term "traditional culture". This was an incorrect description and we have removed and modified it in the manuscript, see page 4, lines 64-71 of the revised manuscript.

Regarding the second question, as you mentioned, the standard method for virus diagnosis is qPCR. However, in our clinical experience, few viral detection methods are routinely used in Chinese hospitals.

While PCR is capable of detecting common pathogens, its positivity rate in CSF is not ideal, especially when dealing with low levels of viral genetic material. Moreover, obtaining a positive result using qPCR assays may require trying multiple specific pathogen detection kits. In contrast, mNGS is a high-throughput sequencing technology that does not require a prior assumption of the pathogen and can simultaneously detect all nucleic acid sequences in a sample, particularly in cases of complex infections or unknown pathogens. Therefore, it can expedite the identification of the causative pathogen (3). Several studies have shown that mNGS can detect pathogens that were identified as negative using conventional methods (including PCR), demonstrating higher sensitivity compared to conventional methods (4-6). We have added more relevant literature to support this statement in the revised manuscript.

Minor issues:

Throughout the abstract there is spaces missing between words. Please correct – there is a few outstanding.

Response: Thank you for pointing out our oversight. We apologize for any errors that were not warranted, and we have carefully reviewed the manuscript to ensure that such oversights do not occur again.

REFERENCES

1. WL Z, YH W, HF L, SY L, SY F, HL W, YJ L, YL L, J H, WC Z, Y Z, GL L, XD Q, HT R, YC Z, B P, LY C, HZ G. 2018. Clinical experience and next-generation sequencing analysis of encephalitis caused by pseudorabies virus. *Natl Med J China* 98:1152-1157.
2. Wang LP, Yuan Y, Liu YL, Lu QB, Shi LS, Ren X, Zhou SX, Zhang HY, Zhang XA, Wang X, Wang YF, Lin SH, Zhang CH, Geng MJ, Li J, Zhao SW, Yi ZG, Chen X, Yang ZS, Meng L, Wang XH, Cui AL, Lai SJ, Liu MY, Zhu YL, Xu WB, Chen Y, Yuan ZH, Li MF, Huang LY, Jing HQ, Li ZJ, Liu W, Fang LQ, Wu JG, Hay SI, Yang WZ, Gao GF, Chinese Centers for Disease C, Prevention Etiology of Acute M, Encephalitis Surveillance Study T. 2022. Etiological and epidemiological features of acute meningitis or encephalitis in China: a nationwide active

- surveillance study. *Lancet Reg Health West Pac* 20:100361.
3. W G, S M, CY C. 2019. Clinical Metagenomic Next-Generation Sequencing for Pathogen Detection. *Annual review of pathology* 14:319-338.
 4. Xing XW, Zhang JT, Ma YB, He MW, Yao GE, Wang W, Qi XK, Chen XY, Wu L, Wang XL, Huang YH, Du J, Wang HF, Wang RF, Yang F, Yu SY. 2020. Metagenomic Next-Generation Sequencing for Diagnosis of Infectious Encephalitis and Meningitis: A Large, Prospective Case Series of 213 Patients. *Front Cell Infect Microbiol* 10:88.
 5. Piantadosi A, Mukerji SS, Ye S, Leone MJ, Freimark LM, Park D, Adams G, Lemieux J, Kanjilal S, Solomon IH, Ahmed AA, Goldstein R, Ganesh V, Ostrem B, Cummins KC, Thon JM, Kinsella CM, Rosenberg E, Frosch MP, Goldberg MB, Cho TA, Sabeti P. 2021. Enhanced Virus Detection and Metagenomic Sequencing in Patients with Meningitis and Encephalitis. *mBio* 12:e0114321.
 6. Wilson MR, Naccache SN, Samayoa E, Biagtan M, Bashir H, Yu G, Salamat SM, Somasekar S, Federman S, Miller S, Sokolic R, Garabedian E, Candotti F, Buckley RH, Reed KD, Meyer TL, Seroogy CM, Galloway R, Henderson SL, Gern JE, DeRisi JL, Chiu CY. 2014. Actionable diagnosis of neuroleptospirosis by next-generation sequencing. *N Engl J Med* 370:2408-17.

Re: Spectrum02264-23R2 (**Metagenomic next-generation sequencing of cell-free DNA for the identification of viruses causing central nervous system infections**)

Dear Dr. Ding Liu:

Your manuscript has been accepted, and I am forwarding it to the ASM production staff for publication. Your paper will first be checked to make sure all elements meet the technical requirements. ASM staff will contact you if anything needs to be revised before copyediting and production can begin. Otherwise, you will be notified when your proofs are ready to be viewed.

Sincerely,
Tulip Jhaveri
Editor
Microbiology Spectrum